Uncovering the Grinnellian niche space of the cryptic species complex Gammarus roeselii

Kabus Jana 1 kabus@bio.uni-frankfurt.de
Cunze Sarah 2
Dombrowski Andrea 1
Karaouzas Ioannis 3
Shumka Spase 4
Jourdan Jonas 1
1 Department Aquatic Ecotoxicology, Johann Wolfgang Goethe Universität Frankfurt am Main , Frankfurt am Main , Germany
2 Department of Integrative Parasitology and Zoophysiology, Johann Wolfgang Goethe Universität Frankfurt am Main , Frankfurt am Main , Germany
3 Institute of Marine Biological Resources and Inland Waters, Hellenic Centre for Marine Research , Anavyssos , Greece
4 Faculty of Biotechnology and Food, Agricultural University of Tirana , Tirana , Albania
Sunny Armando
Electronic publication date: 2023 Aug 3
Publication date: 2023
Volume: 11
Electronic Location ID: e15800
Received 2023 Apr 9; Accepted 2023 Jul 5
Copyright: © 2023 Kabus et al.
Copyright year: 2023
Copyright holder: Kabus et al.
License: This is an open access article distributed under the terms of the Creative Commons Attribution License, which permits unrestricted use, distribution, reproduction and adaptation in any medium and for any purpose provided that it is properly attributed. For attribution, the original author(s), title, publication source (PeerJ) and either DOI or URL of the article must be cited.
License URL: https://creativecommons.org/licenses/by/4.0/

Keywords: Grinnellian niche, Cryptic species, Amphipoda, Niche conservatism, Gammarus roeselii, Balkan peninsula

Funding: Deutsche Forschungsgemeinschaft JO 1465/1-1 This work was supported via funding by the Deutsche Forschungsgemeinschaft to Jonas Jourdan (JO 1465/1-1). The funders had no role in study design, data collection and analysis, decision to publish, or preparation of the manuscript.

==============================
Background

The discovery of cryptic species complexes within morphologically established species comes with challenges in the classification and handling of these species. We hardly know to what extent species within a species complex differ ecologically. Such knowledge is essential to assess the vulnerability of individual genetic lineages in the face of global change. The abiotic conditions, i.e., the Grinnellian niche that a genetic lineage colonizes, provides insights into how diverse the ecological requirements of each evolutionary lineage are within a species complex.

Material and Methods

We sampled the cryptic species complex of the amphipod Gammarus roeselii from Central Germany to Greece and identified genetic lineages based on cytochrome c oxidase subunit I (COI) barcoding. At the same time, we recorded various abiotic parameters and local pollution parameters using a series of in vitro assays to then characterize the Grinnellian niches of the morphospecies (i.e., Gammarus roeselii sensu lato) as well as each genetic lineage. Local pollution can be a significant factor explaining current and future distributions in times of increasing production and release of chemicals into surface waters.

Results

We identified five spatially structured genetic lineages in our dataset that differed to varying degrees in their Grinnellian niche. In some cases, the niches were very similar despite the geographical separation of lineages, supporting the hypothesis of niche conservatism while being allopatrically separated. In other cases, we found a small niche that was clearly different from those of other genetic lineages.

Conclusion

The variable niches and overlaps of different dimensions make the G. roeselii species complex a promising model system to further study ecological, phenotypic and functional differentiation within this species complex. In general, our results show that the Grinnellian niches of genetically distinct molecular operational taxonomic units (MOTUs) within a cryptic species complex can differ significantly between each other, calling for closer inspection of cryptic species in a conservational and biodiversity context.

Introduction

The emergence and routine use of molecular methods throughout the biological sciences revealed more and more species that we consider cryptic. Cryptic species describe species that cannot be distinguished on the basis of morphological characteristics, but are actually separate and distinct evolutionary units within a morphologically described species (Bickford et al., 2007). It is now obvious that cryptic species are a common phenomenon throughout the whole animal kingdom and can be found in aquatic and terrestrial habitats, within invertebrate and vertebrate, in the arctic as well as the deep sea (Hebert et al., 2004; Bickford et al., 2007; Fišer, Robinson & Malard, 2018). These species lack a proper taxonomical description and are therefore classified as molecular operational taxonomic units (MOTUs) based on sequence similarity threshold. The identification of MOTUS is usually based on markers sequencing a short, standardized region of DNA, such as the mitochondrial cytochrome oxidase subunit I (COI) gene (Hebert et al., 2003; Kress et al., 2015). These markers proved to be reliable for species delimitation, as has been demonstrated in several case studies on arthropods (Gibson et al., 2014; Elbrecht et al., 2019; Wattier et al., 2020). Although disadvantages such as putative overestimation, incongruence with nuclear markers (Mamos et al., 2021; Eme et al., 2018; Hupalo et al., 2022) and issues with establishment of species threshold (Lagrue et al., 2014) are known, species identification by COI barcoding has become widely established tool due to its quick and easy applicability.

One assumption as to why cryptic species have not phenotypically differentiated from each other is that their ecological resources do not diverge far enough that a morphological differentiation is necessary, i.e., similar selection pressure has maintained a similar morphology and thus possibly a similar niche (Fišer, Robinson & Malard, 2018). The study of ecological niches can provide insights into the evolutionary history of species (Holt, 2009; Chase & Leibold, 2009). The ecological niche refers to the specific role that a species plays in an ecosystem, including all the biotic and abiotic factors that influence its survival and reproduction. It encompasses a wide range of factors, including biotic interactions such as competition or predation, as well as abiotic factors (Grinnell, 1917; Soberón, 2007). The Grinnellian niche is a subset of the ecological niche space that specifically focuses on the abiotic conditions that a species requires to survive and reproduce (Grinnell, 1917; James et al., 1984; Whittaker, Levin & Root, 1973; Soberón, 2007). In other words, Grinnellian niches refer to the unique and specific combination of resources and environmental conditions that a species requires (Wiens & Graham, 2005). This includes abiotic factors such as temperature, water conditions, or substrate type, which have a direct impact on the distribution and abundance of a species. Generalist species have a larger Grinnellian niche and a higher probability to withstand environmental changes compared to species with a small more specialized niche (Peterson, 2003). The degree of niche conservatism or differentiation between species can indicate how species have evolved over time and how they have adapted to different environments (Wiens & Graham, 2005).

A neglected factor in the characterization of the Grinnellian niche is environmental pollution. Environmental pollution, often of anthropogenic origin, is an indispensable factor that significantly shapes our ecosystems in the densely populated regions of the world (Bernhardt, Rosi & Gessner, 2017; Persson et al., 2022). From a conservation perspective, it is therefore essential to identify the thresholds at which species occur or disappear (Warren, Glor & Turelli, 2010; Peterson, 2011; Broennimann et al., 2012). By comparing the Grinnellian niches of species we can identify the factors that are driving niche differentiation. Accounting for pollution variables in the niches enables to evaluate the anthropogenic impact on species resilience.

In the last two decades, technical achievements and freely accessible databases on climate data resulted in the emergence of numerous studies on niche modelling (Warren, Glor & Turelli, 2008; Peterson, 2011). However, one major challenge is often not yet addressed: In most cases modelling approaches are applied to species identified by morphological rather than molecular differences. This leaves the large part of cryptic biological diversity unconsidered (Bickford et al., 2007; Fišer, Robinson & Malard, 2018 Jourdan et al., 2023). The fact that we have characterised niches mainly for morphologically determined species so far bears the risk of overestimating the sensitivity to environmental conditions. The ecological requirements of different MOTUs are rarely studied, and thus there is a risk that many species that are overlooked cryptic species complexes, are not widespread ecological generalists, but may be small-scale specialized taxa as it is the case for example in cryptic species complexes such bats of the Pipistrellus pipistrellus complex (Sattler et al., 2007), salamanders of the Aneides flavipunctatus complex (Rissler & Apodaca, 2007) or amphipods of the Gammarus fossarum complex (Eisenring et al., 2016; Wattier et al., 2020). These small-scale specialists represent distinct evolutionary pathways and each species may have unique ecological requirements, hence respond differently to environmental changes. The loss of these specialists might not even be detected until it is too late (Niemiller et al., 2013). It is thus important to characterize their ecological niches to identify genetic lineages that may be particularly vulnerable.

A species complex that is well studied concerning the cryptic species status is the freshwater amphipod Gammarus roeselii Gervais, 1835. Grabowski et al. (2017) described the species complex in 2017 and were able to identify at least 13 cryptic species, using a time-calibrated phylogeny and phylogeographic modelling that resulted in a diversification hot spot located in the Balkan Peninsula (Fig. 1). Like many other amphipods, G. roeselii sensu lato is expected to play a key role as a leaf shredder in the nutrient circle (Jourdan et al., 2016) and can occur in high densities (Karaman & Pinkster, 1977). Both in Central Europe and in the Balkans, G. roeselii sensu lato colonize different habitat types (e.g., lakes, mountain streams, rivers) and can also occur in habitats that are heavily anthropogenically stressed (Jourdan et al., 2019; Kochmann et al., 2023). So far, G. roeselii sensu lato is considered to have a high tolerance to many environmental conditions and pollution (Gergs, Schlag & Rothhaupt, 2013; Jourdan et al., 2019). Whether this information can be generalised to all genetic lineages of the species complex remains unclear. Most of the studies are from central Europe—where only MOTU C occurs—and results that have been attributed to G. roeselii might in fact only account for this single MOTU C (Karaman & Pinkster, 1977; Grabowski et al., 2017) (Fig. 1). This is especially crucial for ecotoxicological data, as studies indicate that cryptic species can differ in their tolerance to pollutants (Feckler et al., 2012; Major et al., 2013; Soucek et al., 2013; Feckler et al., 2014; Monteiro et al., 2018 Jourdan et al., 2023). This raises the question if there actually are differences in the Grinnellian niches between the cryptic species and if so, if this can influence the persistence in a changing environment.

Figure 1 Map of the known occurences of Gammarus roeselii in Europe.

Occurrence information was extracted from GBIF (2023), Grabowski et al. (2017) and Csapo et al. (2020). Known MOTUs are represented by coloured circles. Black dots are G. roeselii locations with unknown genetic identity. While MOTU C is widely distributed across Europe, the Balkan region is a diversity hotspot, with many endemic MOTUs. The maps were created using ArcGIS Desktop v.10.8.

A total of 15 years after the first large-scale survey and characterization of the G. roeselii species complex, we conducted another sampling campaign in the manner of (Grabowski et al., 2017), expecting (1) to confirm the molecular patterns and geographical distribution of the G. roeselii species complex identified by Grabowski et al. (2017). We further measured numerous abiotic parameters and conducted several in vitro bioassays to capture local pollution loads. Based on this information we aimed to link the occurrence of MOTUs to local environmental conditions and accordingly describe the Grinnellian niche for different MOTUs. We expected that (2) the Grinnellian niches of the geographically separated MOTUs differ between each other and that (3) the niches differ more the more the MOTUs are phylogenetically distant from each other.

Material and methods

Sampling

We re-sampled sites studied by Grabowski et al. (2017), in order to investigate the stability of local populations of the G. roeselii species complex. In addition, to get an impression about small-scale distribution of each genetic lineage, we further sampled nearby rivers. Furthermore, we also considered additional sample sites in the more recently colonized central Germany (Jourdan et al., 2019; Weigand et al., 2020). Exact coordinates of the 42 sampling sites and their corresponding reference to Grabowski et al. (2017) is given in Table S1. Sampling took place during a field campaign in September 2021. Individuals were sampled by a multi-habitat kick-sampling using hand nets (Bioform V2A; mesh size 500 µm). We preserved all captured individuals in 96% ethanol and stored them at ~10 °C until they were processed in the laboratory for molecular analyses. All sampling was done in close cooperation with local partners; the species are not subject to any protection status. Permission was granted by the Hellenic Ministry of Environment and Energy (Permission No: YПEN/ΔΔΔ/7316/280) and the Albanian National Agency of Protected Areas (Permission No: 194, issued on 16.02.2021). Maps were created using ArcGIS Desktop (v10.8; Esri, Redlands, CA, USA).

Molecular methods

We performed DNA extraction on two individuals of each sampling site using the standard protocol for human or animal tissue and cultured cells of NuceloSpin® Tissue Kit (Machery-Nagel GmbH, Düren, Germany). From each individual two to three pereiopods of one side were carefully removed and placed in a 1.5 ml reaction tube. A total of 25 µL Proteinase K and 180 µL Buffer T1 was added to the sample and they were incubated for 4 h at 56 °C. The lysis, binding and washing of the DNA was carried out as stated in the protocol for human or animal tissue and cultured cells of NuceloSpin® Tissue Kit (Machery-Nagel GmbH, Düren, Germany) for the “High yield, high concentration”-approach.

To identify the different MOTUs cytochrome c oxidase subunit I (COI) gene fragments were amplified using polymerase chain reaction (PCR) with three different primer pairs depending on the MOTU evaluated by Grabowski et al. (2017). Primers used were LCO1490 and HCO 2198 (Folmer et al., 1994), UCOIF and UCOIR (Costa et al., 2009) and COIGrF and COIGrR2 (Grabowski et al., 2017). The full list of primers per individual used is presented in the Table S2. We performed PCR in 30 µL volume, containing 0.4 µL BSA, 3.6 µL MgCl2 (25 mM), 3.0 µL 10 × Taq-Buffer (BioLabs, Boston, MA, USA), 2.4 µL dNTPs (40 mM), 0.4 µL Taq Polymerase (Biolabs, Boston, MA, USA), each primer with a volume of 1 µL (10 mM), 14.2 µL of ddH2O and 4 µL DNA template. Amplification of a 650-bp fragment of the COI gene was carried out using the (PCR) conditions stated in Mamos et al. (2014): Initial denaturation at 94 °C (3 min), 35 cycles of denaturation at 94 °C (20 s), annealing at 50 °C (45 s) and elongation at 65 °C (1 min) and a final extension at 65 °C (2 min). PCR products were purified using the protocol of the PCR clean-up kit of NuceloSpin® (Machery & Nagel GmbH, Düren, Germany). Successful PCR amplification was verified using an aliquot of 5 µL in a GelRed® (VWR) 1.0% agarose gel. The one-sided forward sequencing of the PCR products was performed at Eurofins Genomics’ GATC service “LightRun Tube” (Eurofins Genomics Germany GmbH, Ebersberg, Germany). All sequences were checked using the BLASTn search in GenBank (Altschul et al., 1990). Site 14 did not contain G. roeselii and is therefore not considered in the results. The remaining sequences were then assembled with sequences published in Grabowski et al. (2017) for each MOTU and trimmed to a length of 522 bp in MEGA X (Kumar et al., 2018; Ver. 10.2.6). Alignment was performed with ClustalW (Larkin et al., 2007). With a neighbour-joining tree the 82 sequences were assigned to the MOTUs according to the MOTUs published in Grabowski et al. (2017) (see Fig. S1 and Table S3). We calculated the K2P (Kimura two-parameter) distance as a measure of genetic distance (Kimura, 1980) between and within the MOTUs using MEGA X (Kumar et al., 2018; Tables S4 and S5, Fig. S2). The newly acquired COI sequences are available online on the Barcode of Life Data System (BOLD) via their assigned IDs, provided in Table S2.

Environmental and climatic parameters

Freshwater ecological conditions along the latitudinal gradient from the mountainous Continental Germany to Mediterranean Greece differ significantly (Tockner et al., 2022). In an attempt to capture the variation in abiotic conditions, we measured on site flow velocity (P670; Dostmann electronic, Wertheim, Germany), pH (PHC201; Hach HQ40d multi, Loveland, CO, USA), oxygen content and saturation (LDO101; Hach HQ40d multi, Loveland, CO, USA), as well as conductivity (CDC401; Hach HQ40d multi, Loveland, CO, USA). We extracted the altitude of the sampling sites from Google Earth (http://earth.google.com/). For evaluating the quality of the sampling site sediment and water samples were taken. The water samples were analysed for their concentrations of nitrate (NO3−), nitrite (NO2−) and phosphate (PO43−) with a photometer (FinwellPro for ponds, Glen Iris, Victoria, Australia; MDE GmbH & Co. KG, Puchheim, Germany) and its according measuring kits. Carbonate hardness (HCO3−) was measured using a colorimetric kit (Merck MColortests) (for the environmental data, see Table S6).

We further gained information on hydro-environmental watershed and river characteristics from Domisch, Amatulli & Jetz (2015). This dataset contains spatially continuous and freshwater-specific set of environmental variables for a standardized 1 km river network grid. Domisch, Amatulli & Jetz (2015) used upstream accumulation techniques to obtain watershed contributions for each pixel, providing information on land cover, climatic, river topographic and geological conditions of the years 1970 to 2000. For our modelling approach, we considered flow length (size of the catchment upstream of the sampling site as the sum of 1 km contributing grid cells), cultivated and managed vegetation (lc_avg_07; “cultivated land cover”), urban/built-up land cover (lc_avg_09; “urban land cover”), annual mean temperature (Bioclim1; “annu. mean temperature”) and annual precipitation (Bioclim12; “annu. precipitation”). These variables are computed for a 1 km grid cell of a terrestrial dataset, where the mean of each variables is weighted over the distance of the upstream catchment, starting from the corresponding sampling site (Domisch, Amatulli & Jetz, 2015). This allows us to account for environmental factors in the upstream watershed that have a significant influence on physicochemical conditions of the respective sampling site (for the climatic data, see Table S7 in the Supplementary).

Effect-based assessment of chemical contamination at sampling sites

It was shown that when taking human influences into account, niche differences within a single taxon can be predicted more accurately as seen for example in parakeet distributions (Strubbe et al., 2015). To determine the extent of contamination at each sampling site, we performed a set of in vitro assays to evaluate the baseline toxicity and endocrine activity on the sediment samples. Sediment acts as a sink for various environmental pollutants such as hydrophobic organic contaminants, accumulating and storing pollutants over time (Keiter et al., 2006). As a result, sediment can provide a record of historical contamination levels in a particular location. Specifically, endocrine disruption caused by environmental estrogens can be responsible for disrupting the reproduction and development of organisms, thus a screening for endocrine activity can evaluate the hormonal burden within the aquatic environment (Bergman et al., 2013). As benthic dwellers amphipods are exposed to sediment extensively and sediment analysis is a good indicator of actual exposure to contaminants (Nguyen, Muyssen & Janssen, 2012). In order to consider this pollution information in our analysis, we took sediment samples at each sampling site and stored them at 4 °C until processing. In the laboratory, we then used yeast reporter gene assays to screen for endocrine activity (YES) at the human estrogen receptor α (hERα; Routledge & Sumpter, 1996) and dioxin-like activity (YDS) at the aryl-hydrocarbon receptor agonists (AhR; Stalter et al., 2011) according to Giebner et al. (2018). For the analysis, we eluted 10 g of fine sediment (< 20 µm) in 50 mL methanol an placed the sample on an orbital shaker for 1 h at 220 rpm (GFL 3017; GFL Gesellschaft für Labortechnik GmbH, Burgwedel, Germany). Afterwards, we used an ultrasonic bath (Sonorex RK 52 H; Bandelin electronic, Berlin, Germany) for 10 min to completely detach remaining substances from the sediment. After centrifugation at 4400 rpm for 5 min the supernatant was decanted using a 1.5 µm glassfilter to withhold all suspended soils. The methanol was then removed with a rotary evaporator for 5 min at 56 °C (Heidolph Laborota 4000-efficient, vacubrand CVC 2000; Heidolph Instruments GmbH & Co. KG, Schwabach, Germany; VWR RC-10 Digital Chiller, VWR International GmbH, Darmstadt, Germany) and the remains were dissolved in 500 µL DMSO. Initially, we also tested for androgenic activities at the androgen receptor hAR (Sohoni & Sumpter, 1998) using a YAS assay, but could not find any significant loads in the sediment, so we did not include this in the further analysis. A microtox assay (labelled as “base. tox”) with the bacterium Aliivibrio fischeri was carried out to test for non-specific baseline toxicity of the sediment. A. fischeri is a Gram-negative marine bacterium that produces light through bioluminescence. This bacterium has a high sensitivity to toxic chemicals, making it a useful indicator of acute toxicity. The EC50 value in the microtox assay is the concentration of the sediment that causes a 50% reduction in bioluminescence of the bacteria (i.e., the lower the EC50 value, the more toxic the sample is considered to be). In the further calculations we subtracted the EC50 value by 100 to have high values labelled as more toxic than low values. The microtox assay was in accordance to the ISO guideline 11348-3 (International Organization for Standardization, 1998) and adapted after Escher et al. (2008) for a 96 well plate format. All analysed data is available in the Tables S6 and S7.

PCA and niche-modelling

To compare the Grinnellian niche space of G. roeselii sensu lato and that of the individual MOTUs, we used all the environmental data we collected in combination with online available climatic data to evaluate niches for each MOTU. To do this, we first conducted a principal component analysis (PCA) using the function ‘dudi.pca’ within the R package “ecospat” (Di Cola et al., 2017) using the environmental, chemical and climatic factors (see “Environmental and climatic parameters” and “Effect-based assessment of chemical contamination at sampling sites”). All data was standardised prior performing the PCA so the mean value for all variables is at zero and has a unit variance value by setting “scale = TRUE” within the ‘dudi.pca’ function (Di Cola et al., 2017). Based on a PCA with all the environmental variables, we visualised the niches using the R package “ecospat” (Di Cola et al., 2017). The niche visualisation was based on the first two principal components (PCs) which explained 39% of the variance of the environmental variables. The G. roeselii niche space was then plotted underneath the environmental PCA as a representation of the overall environmental conditions that are suitable for G. roeselii. Overall, the G. roeselii niche space was more or less evenly distributed along all environmental axes. The first two PCs mainly represented a gradient in conductivity, carbonate hardness (HCO3−), annual mean temperature, precipitation and flow length (PC1, explained = 24.1%) as well as a gradient in pH, altitude and mean flow velocity (PC2, explained = 14.9%; see Table S8).

To quantify the niche overlap between each MOTU, the probabilities were calculated where only the first MOTU (i.e., ‘unfilling’), both MOTUs (i.e., ‘stability’) or only the second MOTU (i.e., ‘expansion’) occurs. All MOTUs were compared with each other and displayed in a cross-table. We calculated Schoener’s D (as a measure for niche overlap; Broennimann et al., 2012) and Warren’s I (as a degree of niche similarity of two niches; Warren, Glor & Turelli, 2008) for each MOTU combination. Schoener’s D and Warren’s I range from 0 to 1, with 0 indicating complete dissimilarity and 1 indicating complete similarity. In addition, the p-value of these statistics were calculated using the equivalency test with the hypothesis of a niche overlap that is less equivalent/similar to a randomly generated niche (overlap.alternative = “lower”) and niche conservatism is assumed (expansion.alternative = “lower”, stability.alternative = “higher” and unfilling.alternative = “lower”). We repeated the same analysis with PCs 3 and 4 (see Figs. S3 and S4) that explain additional 11.6% and 9.7% respectively. PC3 and PC4 mainly describe the effect of agriculture, with cultivated land use and O2 saturation (PC 3; explained variance = 11.6%) as well as Nitrate (NO3−) and Nitrite (NO2−) (PC 4; explained variance = 9.7%; see Table S8).

Results

Distribution of the MOTUs sampled

Our COI barcoding approach confirmed the presence of (at least) five MOTUs in our study region (Fig. 2). Most of the sequences acquired at the sampling sites selected accordingly to Grabowski et al. (2017) have the same MOTU classification except for two sampling sites which are our sampling sites 19 and 11 (i.e., ID 13 and 11 respectively in Grabowski et al., 2017). Probably due to our sample size of n = 2, we could not detect co-occurring MOTU E or J here. In Slovenia and Germany we only found MOTU C (sites 1–9). MOTU G (sites 10–21) was found in many stream systems in Albania, as well as in lakes Ohrid and Prespa (Albania and Greece). We also found MOTU GG (site 13 and 15), but we have refrained from including it in the niche analysis, as this MOTU is nestled within MOTU G so we treated this as a MOTU G. In Greece, we found a clear north-south structuring, with MOTU A (sites 22–29) inhabiting the northern part and MOTUs K (sites 35–40) and L (sites 30–33 and 42) the central part of Greece. The occurrence of the MOTUs is therefore in accordance to Grabowski et al. (2017) with the exceptions for sampling site 19 and 11 where only MOTU G was found.

Figure 2 Map of sampling sites and the MOTUs evaluated by the COI barcoding.

MOTUs are named following the classification of Grabowski et al. (2017). Numbers represent the sampling IDs seen in Table S1. (A) All sampling sites in Germany with only MOTU C present. (B) Map of sampling sites in Slovenia, also only MOTU C present. (C) Sampling sites in Albania and Greece with MOTUs G, A, L and K. The maps were created using ArcGIS Desktop v.10.8.

Characterizing the G. roeselii sensu lato niche space

We found G. roeselii sensu lato mainly in larger rivers, but also occasionally in first-order streams. Accordingly, environmental conditions were highly variable, where e.g., altitude varied from sea level up to 905 m, conductivity from 210 to 1100 µS/cm and pH from 7.53 to 9.76 (for a complete list of environmental conditions, see Tables S7 and S8). Pollution load of the sediment also varied strongly, from near-pristine rivers in Slovenia with low nitrate, nitrite and phosphate, to rivers in Greece, characterized by high levels of nitrate and nitrite (100 and 1.64 mg/L respectively; see Table S6). The niche space depicted by the first two PCs was mostly influenced by the conductivity, flow length and annual precipitation along PC1 and the altitude and pH along PC2 (Fig. 3). Pollution variables “baseline toxicity”, “dioxin-like activity” and “estrogenic activity” correlated positively along PC1 and PC2 with dioxin-like activity having a relatively strong impact (Table S8). Nitrate and nitrite concentrations correlated with the occurrence of cultivated land.

Figure 3 Principal Component Analysis (PCA) using PC1 and PC2 with all considered environmental, chemical and climatic variables for all sampled individuals.

Standardized PC1 explains 24.1% and standardized PC2 explains 14.9% of variance. The light grey area represents the Grinnellian niche space of G. roeselii sensu lato of our acquired variables. Dark grey is the area in which 75% of the data is gathered. The lengths of the arrows indicate loadings on the principal components. Blue arrows = climatic and geographical variables, red arrows = pollution variables.

Niche overlap of the MOTU niches

We used the multidimensional PCA information to transform the direct niche space into niches for each MOTU (Fig. 4). The MOTU-specific niches always cover only part of the G. roeselii sensu lato niche. MOTUs C, G and A each have a broad niche, with G and A showing the highest niche overlap and similarity (D and I; Fig. 4). MOTU C and G, both with broad niches but distributed along different PCs, have some significant overlap (D = 0.20; p = 0.0495) and similarities (I = 0.26; p = 0.0495). MOTU L has the smallest niche of all sampled MOTUs, showing no overlap or similarity with niches of MOTU C, G and A (D and I; p < 0.01). The data acquired for MOTU L represents a niche that is only present within the niche of MOTU K (unfilling = 0.00). Additionally compared to MOTU K, there is only a small overlap (D = 0.16; n.s.) and the similarity is modest (I = 0.38; n.s.). Therefore the niche of MOTU L and K can’t be separated by statistical means. MOTU L and A on the other hand show significantly no overlap or similarity (D = 0.00; I = 0.02; both p < 0.01) and the niches of MOTU L and A are therefore significantly different from each other. The southernmost MOTU K and northern most MOTU C show a low overlap but the niches have some significant similarities (I = 0.26; p = 0.0396).

Figure 4 Grinnellian MOTU niches for each MOTU combination.

The niches are spanned along the PC1 and PC2 analysed by the PCA. The outline in each niche plot represents the sensu lato niche space of all analysed data points whereas the dotted inner line is the area in which 75% of the datapoints are situated. The colour of the niches correspond to the according MOTU. Blue coloration is the proportion of overlap where both MOTUs are present. Each marker within the corresponding MOTU represents a sampling site with all its collected data. Density of the occurrence of the second MOTU is illustrated by the shaded area. Calculations are always correspondent to the first MOTU taken into account as explained in the Material and Methods section (“PCA and Niche-Modelling”). Mean for all MOTU niches: I = 0.2626 and D = 0.1787. Asterisks represent the significance level (*p < 0.05; **p < 0.01; no asterisk = not significant).

When considering further environmental variables (i.e., PC 3 and PC 4), we found a less clear differentiation. Here, only MOTU C differed significantly from MOTU A (I = 0.18; Fig. S4). The overlap and similarity (I and D respectively) are high for all MOTUs (mean I = 0.47; mean D = 0.35), indicating that all MOTUs were also found in agricultural dominated regions.

Niche differentiation with genetic distance

The linear regression of the K2P distance and niche overlap (Schoener’s D) did not show a significant correlation (Pearson’s correlation coefficient r = −0.2677; p = 0.4546; Tables S3 and S4, Fig. S2). Thus, we could not find a correlation of increasing niche differentiation and greater genetic distance between MOTUs.

Discussion

Stable distribution pattern

Our samplings confirmed the molecular patterns of the G. roeselii complex sampled by Grabowski et al. (2017) 15 years ago. Moreover, our intensified sampling around the previous sites showed that on a small-scale (e.g., within a catchment) usually only one exclusive MOTU occurs. This is in concordance to most of the findings for Gammarus species in Europe (e.g., G. balcanicus: Mamos et al., 2016, 2021; G. fossarum: Wattier et al., 2020; G. roeselii: Grabowski et al., 2017). Nevertheless, our sample size was small and cannot fully exclude the possibility that co-occurence with rare MOTUs has been overlooked. Co-occurrence of MOTUs is possible as it was shown for other gammarids such as G. fossarum (Copilaş-Ciocianu & Petrusek, 2015; Bystřický et al., 2022) or G. pulex (Grabner et al., 2015). Assuming that MOTUs of a morpho-species use similar resources (e.g., they compete for the same resources, such as food and/or space), competitive exclusion would be expected (Scudo, Ziegler & Lotka, 1978; Tilman, 1982; Webb et al., 2002). If a shift in resource use is unsuccessful, competitive exclusion usually results in the local extinction of one of the species/MOTUs, as the more efficient or better-adapted competitor outcompetes and excludes the other from the area (Tilman, 1982) and a spatial separation of the competing species is evident (e.g., Niphargus hrabei and N. valachicus; Copilaș-Ciocianu et al., 2017; corophiid Amphipods; Borza, Arbačiauskas & Zettler, 2021). Therefore, finding more than one MOTU at the same site seems to be more of an exception than a rule.

MOTU specific niche differentiation

Our niche modelling approach revealed that some MOTUs within the G. roeselii species complex occupied different niches, partially confirming our second hypothesis. MOTU C, G and A all have a broad niche, suggesting that they have a high tolerance to environmental conditions and anthropogenic pollution (Gaston & Spicer, 2001). Interestingly, the wide range of MOTU C is not directly reflected in a larger niche, compared to MOTUs such as A and G. Nevertheless, MOTU C showed a clear differentiation from the other MOTUs (Schoener’s D ≤ 0.2). With its extended geographic distribution, MOTU C is a special case in the G. roeselii species complex. MOTU C is to be assumed the sister clade of MOTU A that separated around 4.5 million years ago (Grabowski et al., 2017). The relatively quick expansion of MOTU C in Central West Europe may have been promoted by human activity in combination with a warming climate (Csapo et al., 2020). The ability to cope with different abiotic conditions is represented by the broad Grinnellian niche, supporting the assumption that MOTU C is an ecological generalist (Csapo et al., 2020). Moreover, the differentiation from other MOTUs suggests that a niche shift has taken place, in favour of colder and larger river systems. This is not surprising, considering the colonization of the vast Danube river system (Sommerwerk et al., 2022) and that of northern river systems such as the Rhine catchment (Wantzen et al., 2022). The possibility to cope with a wide range of environmental conditions due to a broad generalistic niche may be one of the reasons for its widespread distribution in anthropogenically influenced rivers in Central Europe today (Csapo et al., 2020). We also found cases of large niche overlap for example for MOTUs G and A which occur in similar geographic regions. Although these two MOTUs have phylogenetically and geographically separated over 8 Million years ago (Grabowski et al., 2017), they still share similar Grinnellian niches, as commonly found in other allopatrically separated species (Wiens, 2004; Wiens & Graham, 2005). Kozak & Wiens (2006) discussed the role of niche conservatism and allopatric speciation, demonstrated in sister taxa of salamanders Desmognathus and Plethodon. Here, niche conservatism was found even though they only show small climatic differences between the closely related species (Kozak & Wiens, 2006). The strong overlap and highly similar niches of MOTUs A and G in addition to the clearly separated river systems suggest that they have conserved their niche requirements across geographic regions, indicating a degree of niche conservatism. Most likely their ancestral population was lacustrine, but became separated due to geological changes such as the forming of river channels and hydrological networks (Grabowski et al., 2017).

We found the smallest and most clearly separated niche for MOTU L. MOTU L and K belong to the southern members of the G. roeselii species complex, that phylogeographically split from the northern clades about 18 Million years ago (Grabowski et al., 2017). It is assumed that in Greece the southern MOTUs are spatially scattered and only occur in very localised and mostly isolated waterbodies (Grabowski et al., 2017). This is already an indication for possibly smaller niches, simply because amphipods can hardly spread across watersheds and are therefore severely limited in their dispersal capacity. This means that MOTU L colonizes the whole Pinios river system that is shaped by intensive use of water for agriculture, with 95% of available water resource used for crops and farmland (Skoulikidis, 2016). Such river systems are prone to enormous low flows or complete desiccation during summer, making it a system of high human selection pressure (Cooper et al., 2013; Skoulikidis, 2016; Karaouzas et al., 2018). It seems that MOTU L can tolerate this kind of anthropogenic pressure, but further research is needed to determine the specific adaptations or mechanisms that allow it to persist in these challenging conditions (see “Higher tolerance of generalist species in global change?”). In general, when interpreting the niches, it should be considered that the sample size was small. This is mainly due to the large geographical range, the complex analysis of environmental samples and the need for genetic characterization of all organisms. Nevertheless, we were able to sample a representative range of environmental conditions for all MOTUs especially when considering the limited and narrow distribution of most of the MOTUs (Grabowski et al., 2017). Only for the widely distributed MOTU C, which we were only able to collect selectively, we can assume an even greater tolerance to environmental conditions than already shown here.

Overall, we could not find evidence for a relationship between phylogenetic distance and niche differentiation, as shown with the non-significant correlation between K2P distance and Schoener’s D. Thus, our third hypothesis is not supported; it appears more as if the niche differentiation is case-specific, and in some cases the Grinnellian niches of the MOTUs differ in size and space. This was also the case for the amphipod genus Niphargus where different cryptic species showed ecological niche differences but the degree of differentiation was not linked to the genetic distances of the cryptic species (Fišer et al., 2015). The linkage between niche differentiation and phylogenetic relatedness could be more relevant on the clade level as pointed out by Wiens (2008) and Fišer et al. (2015).

In general, it remains to be considered that mitochondrial COI markers reflect geographical diversity well, but in some cases they tend to overestimate species levels in cryptic species and are incongruent with species delimitation by using nuclear markers (Mamos et al., 2021; Hupalo et al., 2022). For future ecological niche modelling of cryptic species it might therefore be useful to further back up the information from COI markers by the additionally use of nuclear markers.

Higher tolerance of generalist species in global change?

The Grinnellian niches represent the abiotic conditions and pollution loads under which we collected the individuals. Actual tolerances may differ. Our initial characterisation of the Grinnellian niche now provides the opportunity for follow-up questions. For example, it remains to be tested whether the small niche of MOTU L may be associated with lower tolerance to the abiotic factors considered here. Vice versa, it can be assumed that MOTU C, G and A tend to have larger tolerances. Such empirical tests of the environmental thresholds found here, would also show whether the species have certain Grinnellian niches only because of geographical barriers, i.e., it can be checked whether the approach used here incorrectly predicts too small Grinnellian niches for dispersal-limited/spatially structured species (e.g., MOTU L).

In recent years, Gammarus spp. have increasingly been used as test organisms for studying adaptation processes to various anthropogenic stressors (Schulz & Liess, 1999; Bundschuh & Schulz, 2011; Shahid et al., 2018a, 2018b; Siddique et al., 2020; Siddique, Shahid & Liess, 2021; Grethlein et al., 2022). So far, mostly intraspecific differences between populations in anthropogenically stressed regions were compared with pristine regions. The G. roeselii species complex provides an excellent framework to study adaptations in five closely related cryptic species that occur at both, the inter- and intraspecific levels.

Niche differences as an indicator of trait divergence?

The pronounced difference in the Grinnellian niches, especially of MOTU L and C, calls for investigation under an integrative framework to include a morphological/phenotypic characterization and focus on trait differentiation. Follow-up studies, should quantify a set of characteristics in MOTU L compared to MOTU A, C and G in order to see the role of ecological influence on the phenotype. If phenotypic differentiation is visible within the cryptic species complex, they should be more pronounced in MOTUs where we have observed niche differentiation (Whitlock, 1996; Fišer, Robinson & Malard, 2018). One might specifically expect MOTU L to have a different spectrum of phenotypic differentiation compared to MOTUs with a broad niche size (Sexton et al., 2017). Whether this phenotypic differentiation might then be more or less pronounced and in correlation with niche size differences remains to be seen. Additionally, if phenotypic differentiation is found, it should be discussed whether these differences also affect functional ecosystem processes (Bickford et al., 2007; Fišer, Robinson & Malard, 2018).

Conclusion

In conclusion, we were able to confirm the molecular patterns identified by Grabowski et al. (2017) for five MOTUs. The Grinnellian niche space of the cryptic species complex of G. roeselii differs from the niches occupied by each MOTU calling for closer inspection of cryptic species in a conservational and biodiversity context. The overlap of some MOTU niches suggests that although allopatrically separated, the niches of the MOTUs are still similar and can be the result of niche conservatism. On the other hand, there is also niche differentiation, which is not unexpected given the small-scale spread of individual MOTUs in the south of the Balkan Peninsula, compared to MOTUs distributed over large parts of Europe. Hence, our results highlight the need to consider cryptic species complexes as such, since individual species must be assumed to have different ecological requirements and tolerances. The G. roeselii species complex offers a test system to gain a deeper understanding of the mechanisms that drive niche differentiation and ecological specialization. An integrative framework that incorporates molecular, morphological and ecological data is essential for this.

Supplemental Information

Supplemental Information 1 Sampling site information with occurring MOTUs evaluated by the COI barcoding.

MOTUs follow the naming of Grabowski et al. (2017). Sampling locations are given in decimal degree. If sampling sites are the same or close (<2 km) to the ones provided in Grabowski et al. (2017) the corresponding ID is supplied. The coordinates were used in the making of the map in Fig. 1. Country abbreviations: AL, Albania; DE, Germany; GR, Greece; SL, Slovenia.

Click here for additional data file.

Supplemental Information 2 mtCOI Primers used for each individual evaluated in the genetic analysis and corresponding BOLD IDs.

Barcode of Life (BOLD) IDs run from CRSEE001-23 to CRSEE0082-23. Primers used are LCO1490 and HCO 2198 (Folmer et al., 1994), UCOIF and UCOIR (Costa et al., 2009) and COIGrF and COIGrR2 (Grabowski et al., 2017) and are further described in the Materials and Methods.

Click here for additional data file.

Supplemental Information 3 Additional mtCOI sequences used for creating a phylogenetic Neighbor-joining tree and delimitation of the MOTUs.

All G. roeselii sequences were published by Grabowski et al. (2017). The sequence of G. fossarum is published in Wattier et al. (2020) and taken as outgroup. All sequences are available online at the National Center of Biotechnology Information (NCBI).

Click here for additional data file.

Supplemental Information 4 Genetic mean distance (K2P) between each MOTU.

The between each MOTU K2P distance is situated in the bottom left and standard error in the upper right corner of the matrix. K2P distance is calculated after Kimura (1980).

Click here for additional data file.

Supplemental Information 5 Genetic distance (K2P) within each MOTU.

K2P distance is calculated within each MOTU and the standard error is given below each value. K2P distance is calculated after Kimura (1980).

Click here for additional data file.

Supplemental Information 6 Environmental and chemical data used for the analysis.

Raw data of each environmental and chemical variable and their values are listed here. The corresponding units are displayed in brackets. All data displayed was used to evaluate the PCA and the niches for each MOTU. If the value of the estrogen or dioxin-like activity was below the limit of quantification (LOQ) it is stated as <LOQ. LOQ for the YES = 0.305 ng/g; LOQ for YDS = 0.092 mg/g. In the analysis it was set to zero to account for no estrogenic or dioxin-like activity.

Click here for additional data file.

Supplemental Information 7 Climatic data attained from Domisch et al. (2015) for sampling sites used.

Raw data of climatic variables and values are listed here. The corresponding units are displayed in brackets. This raw data was attained from Domisch, Amatulli & Jetz (2015).

Click here for additional data file.

Supplemental Information 8 Loadings of each variable used in the PCA with the first six PCs shown.

The loadings of the variables determine the strength of influence on the data distribution. The cumulative explained variance (%) is given below.

Click here for additional data file.

Supplemental Information 9 Neighbour-joining tree of G. roeselii mtCOI sequences acquired in this publication as well as sequences published by Grabowski et al. (2017) and Wattier et al. (2020).

The neighbour-joining tree was calculated using the bootstrap method with 1,000 iterations with the percentage (>70) of replicate trees on each node. MOTU delimitation follows the sequences determined by Grabowski et al. (2017). The evolutionary distances were computed using the Kimura 2-parameter method with the units of the number of base substitions per site. Newly acquired sequences are abbreviated with “Cr-Site number-Individual Number”. The corresponding site numbers and additional sequences used are listed in in the Supplements. G. fossarum is taken as outgroup.

Click here for additional data file.

Supplemental Information 10 Linear regression of genetic K2p distance to Schoener’s D.

K2p distance was calculated in MEGA X and represents the genetic distance between each MOTU. Schoener’s D is taken as a measure for niche overlap. Pearson’s r is 0.2677 and p-value is not significant (p-value = 0.4546) marking the linear regression of the two variables as not significant.

Click here for additional data file.

Supplemental Information 11 Principal Component Analysis (PCA) using PC3 and PC4 with all considered environmental, chemical and climatic variables for all sampled individuals.

Standardized PC3 explains 11.6% and standardized PC4 explains 9.7% of variance. The light grey area represents the Grinnellian niche space of G. roeselii regular lato of our acquired variables. Dark grey is the area in which 75% of the data is gathered. Black dots represent the data for each sampling site. The lengths of the arrows indicate loadings on the principal components. Blue arrows = climatic and geographical variables, red arrows = pollution variables.

Click here for additional data file.

Supplemental Information 12 MOTU niches for each MOTU combination using PC3 and PC4.

The niches are spanned along the PC3 and PC4 analysed by the PCA. The outline in each niche plot represents the sensu lato niche space of all analysed data points whereas the dotted inner line is the area in which 75% of the datapoints are situated. The colour of the niches correspond to the according MOTU. Blue coloration is the proportion of overlap where both MOTUs are present. Each marker within the corresponding MOTU represents a sampling site with all its collected data. Density of the occurrence of the second MOTU is illustrated by the shaded area. Calculations are always correspondent to the first MOTU taken into account as explained in the Material and Methods section (“PCA and Niche-Modelling”). Asterisks represent the significance level (*p < 0.05; **p < 0.01; no asterisk = not significant). For all MOTUs combined: mean I = 0.4694 and mean D = 0.3471).

Click here for additional data file.

We would like to thank Simon Hornung and Lars Pelikan for the great help and company during the fieldwork. Additionally, we would like to express our greatest thanks to M. Grabowski, C. Fišer, C. Nowak, A. Weigand and C. Albrecht who all helped and supported this project. The authors would also like to thank the local people in Albania, Slovenia and Greece for their hospitality and help in gathering all the material.

Additional Information and Declarations

Competing Interests

Author Contributions

Field Study Permissions

DNA Deposition

Data Availability

The authors declare that they have no competing interests.

Jana Kabus conceived and designed the experiments, performed the experiments, analyzed the data, prepared figures and/or tables, authored or reviewed drafts of the article, and approved the final draft.

Sarah Cunze conceived and designed the experiments, performed the experiments, authored or reviewed drafts of the article, and approved the final draft.

Andrea Dombrowski performed the experiments, analyzed the data, authored or reviewed drafts of the article, and approved the final draft.

Ioannis Karaouzas performed the experiments, authored or reviewed drafts of the article, and approved the final draft.

Spase Shumka performed the experiments, authored or reviewed drafts of the article, and approved the final draft.

Jonas Jourdan conceived and designed the experiments, performed the experiments, authored or reviewed drafts of the article, and approved the final draft.

The following information was supplied relating to field study approvals (i.e., approving body and any reference numbers):

Field sampling was permitted by the Hellenic Ministry of Environment and Energy (YПEN/ΔΔΔ/7316/280) and the Albanian National Agency of Protected Areas (NAPA) (Permission No: 194, issued on 16.02.2021).

The following information was supplied regarding the deposition of DNA sequences:

The mtCOI sequences are available at the public Barcode of Life Data system (BOLD): CRSEE001-23 to CRSEE0082-23 (Table S2).

https://www.boldsystems.org/index.php/Public_RecordView?processid=CRSEE001-23

https://www.boldsystems.org/index.php/Public_RecordView?processid=CRSEE082-23

The following information was supplied regarding data availability:

The raw data and additional information are available in the Supplemental Files.

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
