# Peer review of "Uncovering the Grinnellian niche space of the cryptic species complex Gammarus roeselii"

_PeerJ, doi:10.7717/peerj.15800_

## Round 0.1 · original submission · Major Revisions

Dear authors,

After reviewing your work and considering two contrasting reviews, I find your research to be valuable and deserving of publication. However, there are major corrections that need to be made. Specifically, the reviewers suggest improving the introduction, providing better explanations of the materials and methods, and depositing the sequences used in a repository (such as NCBI GenBank or ENA) and making them publicly available in the BOLD dataset mentioned in the text (which is currently unavailable). Additionally, it is important to carefully review the cited literature and confirm the correct number of sampling sites used (whether it is 47, 42, or 41), and add more literature to the introduction and discussion sections. Lastly, the supplementary materials require a thorough review to ensure that all figures and tables have appropriate captions and are easy to understand.

Best regards,

Armando Sunny

Reviewer 1 ·

Basic reporting

This Ms tentatively assess the existence of Grinnelian niche space of the cryptic species complex Gammarus roeselii for a set of abiotic factors, including anthropogenic stressors.

Numerous key papers linking cryptic diversity to ecology (including niche and ecotoxicology) for amphipods are missing. A list is provided below.

Eisenring, M., Altermatt, F., Westram, A. M. & Jokela, J. Habitat requirements and ecological niche of two cryptic amphipod species at landscape and local scales. Ecosphere 7, e01319. (2016).
Eme, D. et al. Do cryptic species matter in macroecology? Sequencing European groundwater crustaceans yields smaller ranges but does not challenge biodiversity determinants. Ecography 41, 424–436 (2018).
Feckler, A. et al. Cryptic species diversity: an overlooked factor in environmental management?. J. Appl. Ecol. 51, 958–967 (2014).
Feckler, A., Thielsch, A., Schwenk, K., Schulz, R. & Bundschuh, M. Differences in the sensitivity among cryptic lineages of the Gammarus fossarum complex. Sci. Total Environ. 439, 158–164 (2012).
Fišer, Ž, Altermatt, F., Zakšek, V., Knapič, T. & Fišer, C. Morphologically cryptic amphipod species are “ecological clones” at regional but not at local scale: a case study of four Niphargus species. PLoS ONE 10, e0134384 (2015).
Galipaud, M., Bollache, L. & Lagrue, C. Variations in infection levels and parasite-induced mortality among sympatric cryptic lineages of native amphipods and a congeneric invasive species: Are native hosts always losing?. Int. J. Parasitol. Parasites Wildl. 6, 439–447 (2017).
Major, K., Soucek, D. J., Giordano, R., Wetzel, M. J. & Soto-Adames, F. The common ecotoxicology laboratory strain of Hyalella azteca is genetically distinct from most wild strains sampled in eastern North America: common lab strain of H. azteca is distinct from wild strains. Environ. Toxicol. Chem. (2013).
Westram, A. M., Baumgartner, C., Keller, I. & Jokela, J. Are cryptic host species also cryptic to parasites? Host specificity and geographical distribution of acanthocephalan parasites infecting freshwater Gammarus. Infect. Genet. Evol. 11, 1083–1090 (2011).
Westram, A. M., Jokela, J. & Keller, I. Hidden biodiversity in an ecologically important freshwater amphipod: Differences ingenetic structure between two cryptic species. PLoS ONE 8, e69576. (2013).

In addition, all the SUP-MAT tables and figures are lacking titles and are therefore hard, not to say impossible, to follow without the Ms aside.

The references list includes more than 50 typos (e.g. full journal name vs abbreviated names, genus and species names not in italics…).

Experimental design

I would recommend rejecting this Ms based on two major and detrimental limitations:

1) The study is associated, at least in the Balkans, to a totally inappropriate sampling size to assess possible co-occurrence of MOTUs at a given site. Grabowski et al 2017, clearly established, despite being itself based on rather limited sampling size, that MOTUs can co-occur in the Balkans. With only two samples per site, the authors are at highly risk to have faulty results for some/many sites. One site claimed to be associated to one MOTUs could be a mixture of MOTUs. The sampling size should, at least, be one order of magnitude higher.

2) Including anthropogenic stressors as reflecting evolutionary pressure that might have been the drivers of cryptic speciation is erroneous. Grabowsky et al 2017 clearly showed, as pointed out by the authors of the present Ms, that the time-frame of diversification is spanning My, therefore not matching Anthropocene.

In addition, many other aspects are on the downside, including, the list not being limitative:

1) Not all the expectations at the end of the aims of the introduction are convincing. Why testing geographic distance and not the type of habitat?
2) As pointed-out in the introduction, G. roeseli may occur in different habitat types (e.g. lakes, mountains streams, rivers). Nowhere in the Ms is specified which type of habitats were sampled.
3) The Ms is inconsistent about the number of sites used: 47, 42 or 41?
4) What is the point of including two sites from Germany? This sampling strategy is awkward and not justified.

Validity of the findings

no comment

Reviewer 2 ·

Basic reporting

The manuscript is ell structured and written in clear English. The results and figures are appropriate and relevant to hypothesis. I read the manuscript with high interest and I thing that it's important contribution fit to be published in peerj.
The introduction could provide more background in the case of cryptic species recognition and identification. Especially since the whole reasoning is based on COI MOTU delimitation, its’ strength should be mentioned (fast/unambiguous indentation in case of poorly known or cryptic animals, e.g. Wattier et al. 2020 and references therein) but also disadvantages that are already detected like putative overestimation and incongruence with nuclear markers (e.g. Mamos et al 2021:Sci Rep 11, 21629) and issues with establishment of species threshold (e.g. Lagrue et al 2014).

Experimental design

Research question and hypothesis are well defined, relevant and interesting. Research experimental design is accurate.
The study shows that G. roeselii provides important study model in evolutionary research related to ecological specialisation.
The investigation is conducted rigorously but some methods could be described with more detail and maybe extended (details below).

Validity of the findings

The study provides important insight into ecological differentiation of model gammarid group that is based on solid data collection.
The molecular data should be deposited in the obligatory data repositories (NCBI GenBank or ENA) and publicly available in the mentioned in text BOLD dataset (not available at the moment of review).
Conclusions are well stated and linked to the hypothesis.

Additional comments

Detailed comments based on numbered lines:

M&M
Line 157 redundant repetition of “sampling”
188-185: was sequencing done both direction or only one?
188: please add reference for BLAST. Were the sequences tested for presence of pseudo-genes (e.g. through transcription)?
197-199 All sequences should also be deposited in GenBank or ENA. At the moment the sequences are not accessible through BOLD.
225 The Tab. S7 and S8 are cited but it seems that Tab. S5 is cited later, please verify if numbering is continuous through the text.
268-270 Did the data standardisation was needed prior PCA analysis (if so how was it done)? Also I am advising providing basic descriptive statistics (mean/min./max/SD) for all variables for better understanding of the data used in the study.
273 39% not 49%
279 I have big concern regarding using the only the first two PCs as they explain less then 50% of the variance. Is there some strong reasoning for this choice (e.g. K-means, elbow test). I would suggest using also PC3 and PC4 as they explain additional 20 % of variance with strong correlation of some variables. For example the PC3 and PC4 could be analysed as another or further niche space (like on Figure 4).
Results:
338-339 “Thus, we could not find a correlation between increasing niche differentiation with greater genetic distance between MOTUs”
Discussion:
343-348 in some specific cases/regions gammarids MOTUs occur in sympatry but in majority of the cases all main three morphospecies complexes from Europe, including cited G. pulex/fossarum and studied G. roeselii show only one MOTU per station both in wide and local scales(e.g. G. fossarum: Wattier et al 2020, G. balcanicus : Mamos et al 2016/2021, G. roeselii Grabowski et. 2017). This is just a suggestion but in my opinion this sentences should be modify to show that that obtained results are in concordance with what is seen for the gammarids in Europe while sympatry is rather exception.
367: consider replacing “large geographic distance” with “wide range”
359-360 consider rephrasing. At the moment the confirmation of second hypothesis is not clear
407-410 maybe it’s worth mentioning that when considering geography, many of the studied MOTU’s have rather narrow distribution and, highly likely, in the current studies, majority of MOTU’s ranges were covered.
In my opinion the chapter is lacking some short discussion regarding the limitation of using only the COI MOTU, that in case of gammarids reflect well geographical diversity, however on nuclear level this diversity is not so clear (e.g. Mamos et al 2021, 28S data in the Grabowski et al 2017) and maybe the niche diversity is better reflected on nuclear level or in patterns of genes expression, for which this study is giving fundaments.

Cited literature:
Lagrue et al. 2014. Freshwater Biology 59, 2555-2570.
Wattier et al. 2020. Sci Rep 10, 16536.
Mamos et al. 2016. MolEcol 25, 795-810/2021. Sci Rep 11, 21629.
Grabowski et al. 2017. PeerJ 5, e3016.

---

## Round 0.2 · accepted · Accept

Dear authors,

I am delighted to inform you that the reviewers have concurred with the implemented corrections, thus deeming the manuscript ready for publication. I extend my heartfelt gratitude to you for choosing PeerJ as a platform to showcase your intriguing work.

Warm regards,

Armando Sunny

Reviewer 2 ·

Basic reporting

no further comments

Experimental design

no comment

Validity of the findings

All the data are provided and publicly available, analysis is robust.

Additional comments

The manuscript was improved rigorously using the provided comments and all the doubts clarified, therefore I don't have any further comments.